# Egocentric Cooking Motion Dataset

Chun-An Chou[1] and Yu-Hui Huang[1]

[1]Department of Electrical Engineering, Yuan Ze University, Taoyuan, Taiwan

*Abstract*— In this paper, we introduce a novel dataset featuring hand pose annotations for video clips sourced from Epic Kitchen, an existing cooking dataset. By augmenting the dataset with this additional annotation, we aim to advance research in the realm of multi-modality for cooking robots developments.

## I. INTRODUCTION

The rise of egocentric or first-person vision has opened up new avenues for understanding human activities and interactions from a user-centric perspective [1]- [5]. As people work on daily tasks, wearable cameras and sensors have the potential to capture invaluable insights into how we perceive and navigate our environment. This egocentric viewpoint is particularly relevant for applications in the domain of assistive robotics, such as cooking support systems.

Researchers have explored the use of egocentric vision to develop intelligent cooking robots that can observe and learn from human behavior during food preparation. By analyzing the hand motions, object manipulations, and contextual cues extracted from egocentric video, these systems aim to build an understanding of the step-by-step cooking process. This knowledge can then be leveraged to provide personalized guidance, automate certain repetitive tasks, and even learn novel cooking techniques through imitation.

Beyond just observing the user, some cooking robots incorporate egocentric sensing to also understand their own physical embodiment and spatial awareness within the kitchen environment. By combining first-person visual inputs with proprioceptive feedback from their actuators, these robotic systems can better coordinate their movements, avoid collisions, and seamlessly interact with the user and cooking utensils. This synergy between egocentric perception and robotic control is crucial for developing intuitive and collaborative cooking assistants.

The advancement in egocentric vision holds significant promise for cooking robots. Integrating egocentric perception can enhance human-robot interaction, streamline food preparation, and improve the overall cooking experience for users in both domestic and commercial settings.

In this work, we introduce a supplementary cooking motion dataset that extends the EPIC-KITCHENS dataset [3]. The goal of our effort is to provide detailed annotations of hand poses for the cooking-related motions captured in the original video. Adhering to the joint labeling conventions defined in [6], we annotate 21 key joint locations for each hand across the cooking activity video clips as shown in Fig. 1.

Together with the motion annotation data we are providing for the EPIC-KITCHENS video clips, our goal is to facilitate novel research at the intersection of natural language processing and 3D motion understanding. By pairing the rich egocentric video footage with detailed annotations of hand poses and joint movements, we hope to empower the development of multimodal learning frameworks that can jointly reason about the linguistic descriptions of cooking actions and the underlying kinematic patterns exhibited by expert chefs.

Such multimodal models could enable a wide range of applications. For example, intelligent cooking assistants could learn to follow step-by-step recipe instructions by simultaneously comprehending the textual descriptions and mimicking the associated hand motions demonstrated in the video data. Conversely, these systems might be able to generate natural language explanations of the correct techniques based on an observed set of cooking motions.

Beyond just cooking, this dataset can serve as a testbed for advancing multimodal perception and reasoning more broadly. Researchers exploring human-robot interaction, activity recognition, and imitation learning will find immense value in the ability to ground language-based knowledge with fine-grained 3D motion data captured from the first-person perspective. We are excited to see how the community will leverage this unique dataset to push the boundaries of what is possible when combining the modalities of vision, kinematics, and natural language.

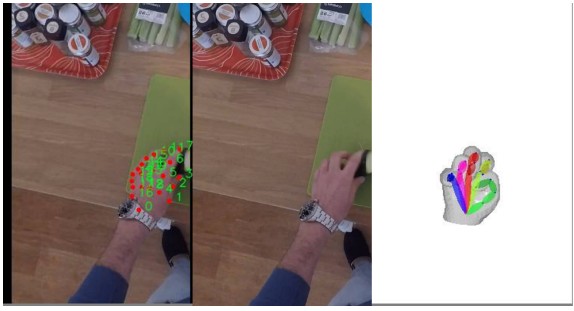

Fig. 1. An example of our hand pose annotations.

## II. CONCLUSIONS

In conclusion, our introduction of an egocentric cooking motion dataset, building upon the foundation of EPIC-KITCHENS, represents a significant step forward in supporting research and development of intelligent cooking assistance systems. We aim to empower the research community to explore new frontiers at the intersection of computer vision, robotics, and natural language processing.

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
