# OpenReview forum: "Egocentric Cooking Motion Dataset"
_IEEE.org/2024/ICRA/Workshop/CookingRobot — CookingRobot2024 Poster_

### Official Review · Reviewer_5Uip · 2024-04-15
**Review of "Egocentric Cooking Motion Dataset"**

**Rating:** 7
**Confidence:** 4

**Review:**

This paper introduces a dataset adding hand pose annotations to existing cooking video datasets, showcasing examples of hand pose annotations on cooking videos, including cutting, frying, and washing.

Comments on Paper:
* This work is well-motivated to add hand pose annotations to an existing cooking dataset, which includes linguistic descriptions. It takes a step forward in addressing the important issue in cooking robotics, creating a dataset that allows for joint reasoning about linguistic descriptions such as recipes and corresponding cooking actions.
* The details of the annotation method are missing. It would be helpful to provide information on any tools or software used for annotation, how it handles ambiguous cases (e.g., occlusion), and any quality control measures implemented to ensure the accuracy and reliability of annotations.
* The authors could argue the cooking-specific challenges in hand pose annotations (e.g., hand pose in relation to tools, etc.) and discuss how they could be possibly addressed in future works on the topic.

Comments on Video:
* A nice video showing hand pose annotations on cooking videos, including cutting, frying, and washing.
* It would be interesting to see failure cases in annotation and explanations of the challenges involved.

---

### Official Review · Reviewer_8ZGv · 2024-04-15
**Review for "Egocentric Cooking Motion Dataset"**

**Rating:** 7
**Confidence:** 4

**Review:**

The paper proposes an egocentric dataset of cooking tasks by annotating the EPIC-KITCHENS dataset. They annotate hand poses on the video, which we can expect to help us learn the motions of robot cooking tasks. Although the paper itself is a proposal of the dataset, adding experiments of application of this dataset will make the proposal more strong, e.g., human-to-robot learning with keypoint imitation learning.

Comments on the video
- Showing the dataset in the video helps us understand how the key points are annotated.